# Burdening and Protective Organisational Factors among International Volunteers in Greek Refugee Camps—A Qualitative Study

**DOI:** 10.3390/ijerph19148599

**Published:** 2022-07-15

**Authors:** Isabel Josam, Sarah Grothe, Daniel Lüdecke, Nico Vonneilich, Olaf von dem Knesebeck

**Affiliations:** Institute of Medical Sociology, University Medical Center Hamburg-Eppendorf, 20246 Hamburg, Germany; sarah.grothe@stud.uke.uni-hamburg.de (S.G.); d.luedecke@uke.de (D.L.); n.vonneilich@uke.de (N.V.); o.knesebeck@uke.de (O.v.d.K.)

**Keywords:** humanitarian aid worker, volunteer, humanitarian aid, qualitative research, organisational factors, organisation, gender, refugee camp, disaster relief

## Abstract

A majority of the workforce in the humanitarian aid consists of volunteers who partly suffer from health problems related to their voluntary service. To date, only a fraction of the current research focuses on this population. The aim of this qualitative explorative study was to identify burdening and protective organisational factors for health and well-being among humanitarian aid volunteers in a Greek refugee camp. To this end, interviews with 22 volunteers were held on site and afterwards analysed by using qualitative content analysis. We focused on international volunteers working in Greece that worked in the provision of food, material goods, emotional support and recreational opportunities. We identified burdening factors, as well as protective factors, in the areas of work procedures, team interactions, organisational support and living arrangements. Gender-specific disadvantages contribute to burdening factors, while joyful experiences are only addressed as protective factors. Additionally, gender-specific aspects in the experience of team interactions and support systems were identified. According to our findings, several possibilities for organisations to protect health and well-being of their volunteers exist. Organisations could adapt organisational structures to the needs of their volunteers and consider gender-specific factors.

## 1. Introduction

The number of people in need of humanitarian aid increased continuously over the past years. In 2020, 243.8 million people required humanitarian assistance [1]. To match this rising demand for humanitarian aid, more people are involved in humanitarian field work [2]. As the need for humanitarian assistance is predicted to continue to grow further in the future [3], so will the humanitarian work force. Therefore, it is important to understand how this work environment affects the people that are engaged in providing humanitarian aid. Humanitarian aid workers (HAWs) are frequently exposed to trauma [4,5] and, additionally, the work is often associated with stressful conditions, such as political instability, ambiguity and high urgency [6,7]. In relation to their work, HAWs experience several adverse mental and physical health outcomes. These include an increased risk of anxiety, post-traumatic stress disorder (PTSD), depression, burnout, general distress, emotional exhaustion, depersonalisation, hazardous alcohol consumption and a reduced overall well-being [8,9,10,11,12,13]. These findings emphasise the importance to avoid or mitigate possible negative impacts of this line of work.

Most studies conducted so far focused primarily on professional HAWs, leaving out of sight the many volunteers working in the humanitarian field. As an example, the International Federation of Red Cross and Red Crescent Societies alone engages almost 15 million volunteers in their operations each year, compared to only 516,000 permanent employees [14]. Considering these numbers, the needs of this group should not be overlooked. In addition, recent studies indicate that volunteers report even higher levels of psychological morbidity than professional HAWs [9,15,16]. The experience of professional HAWs will not readily translate to volunteers, as the initial situation and working conditions differ. For example, frequently discussed stressors in the research among professional HAWs are job insecurity, limited career opportunities and salary [4,7,17,18]. As these topics are not relevant for volunteer workers, it highlights the importance of investigating this group separately. Accordingly, this study focuses particularly on international volunteers in the humanitarian aid that were working in a Greek refugee camp and were engaged in the provision of food, material goods, emotional support and recreational opportunities.

Previous research indicated a number of health-related factors. Demographic characteristics, such as age, number of children or family status, are associated with the mental health of humanitarian volunteers [9,19,20]. Associations of female gender and an increased psychological morbidity were also identified by recent studies [8,9,19]. So far, the reasons for the differences between genders have not been fully disclosed and, thus, need to be further investigated. Organisational aspects can also have a burdening or protective effect. The duration of shifts, length of operation periods and perceived organisational support influence the psychological stress levels and probability for PTSD [20,21]. Additionally, these factors also impact the resilience of volunteers in the humanitarian aid [22].

In order to protect volunteers from negative impacts of their humanitarian engagement, it is important to have a comprehensive overview of all organisational factors that can influence the health and well-being. This offers a direct approach for organisations to positively influence their volunteers and, therefore, the entire organisation. The currently available literature, however, does not allow a complete overview of the influencing organisational factors, as no explorative study in this field has been undertaken so far.

In an attempt to narrow this research gap, this study aims to identify burdening and protective factors for health and well-being within the organisational aspects of the work for international volunteers using an inductive, qualitative approach. To this end, we focus on nongovernmental organisations (NGOs) that are part of the European refugee response in Greece and have been subjected to great stress because of the COVID-19 pandemic and the evolving situation on the Greek islands. In the investigated setting, volunteers play an essential role for the provision of humanitarian aid. Many humanitarian organisations are grassroot organisations that strongly rely on volunteers for their daily operations [23]. More specifically, the following research questions will be addressed: 1. Which burdening factors in the organisational structures can be identified for volunteers in the humanitarian aid? 2. Which protective factors in the organisational structures can be identified for volunteers in the humanitarian aid? 3. Can gender differences be identified in the experiences of organisational structures for volunteers in the humanitarian aid?

## 2. Materials and Methods

### 2.1. Study Site

Since 2015, over 1.1 million refugees arrived in Greece by crossing the Mediterranean Sea and approximately 45% of them entered the country through the island of Lesvos [24]. This put an extreme strain on the small island and led to the emergence of numerous NGOs [23]. Many of these NGOs are grassroot organisations with no prior experience in the humanitarian work. They employ a big variety of volunteers from all over the world for differing amounts of time [23].

The data collection for this study took place between December 2020 and January 2021. This was shortly after a fire destroyed the Reception and Identification Centre Lesvos, which had been the placement for most refugees. To substitute the loss of housing, a temporary camp was set up to house approximately 7300 people [25]. The military controlled the new temporary camp and only a limited number of NGOs had access to work inside the camp.

### 2.2. Study Design and Participants

For this qualitative study, international volunteers were recruited using a purposive sampling approach. Calls to participate were distributed in locally relevant social media channels and messenger services. The digital approach was necessary to comply with local lockdown regulations due to the COVID-19 pandemic. This approach was still deemed suitable because most volunteers were frequent users of these online tools. Eligibility criteria for participants (P.) were a current engagement as an international volunteer in an NGO working inside the refugee camp and at least three weeks of work experience within the NGO on site. Three weeks of experience were deemed sufficient for this study, as most NGOs required a minimum stay of two weeks. Accordingly, many volunteers were also engaged for short-term assignments and their perspective should be included. In response to the announcements, 33 people came forward, of whom 28 met the inclusion criteria. Four eligible participants dropped out before the interviews took place and two did not show up for the interview appointment. A balanced gender ratio was achieved in the sampling process. In total, *n* = 22 people participated in the study, of whom 12 were females and 10 were males. Due to the small and specific population investigated, this sample size was deemed suitable to reach sufficient saturation. The mean age was 35 years (range 23–68 years). People from nine different nationalities participated in the study. Of 22 consenting participants, 6 were from the Netherlands; 3 each were from Belgium, France and the U.K.; 2 each were from Denmark and Italy; and 3 more were from Canada, Germany and Norway. Three participants completed secondary education as their highest level of education, sixteen completed a form of tertiary education and three participants held a doctorate degree. The volunteers worked for eight different NGOs that varied in size and experience. The range of tasks of the NGOs included the provision of food, material goods, emotional support and recreational opportunities. All interviewed volunteers worked inside the Reception and Identification Centre Lesvos and worked on five to six days per week. 

The Local Psychological Ethics Committee at the Centre for Psychosocial Medicine, University Medical Centre Hamburg, approved the study (No. LPEK-0235). Before the interview, each participant was informed about the objective of the study and its voluntary nature. Information on anonymity and confidentiality was given. All participants signed an informed consent form in which they agreed to the terms and the audio recording of the interview. In order to present the findings of this study the consolidated criteria for reporting qualitative research (COREQ) were considered [26].

### 2.3. Qualitative Interviews

Semi-structured interviews were conducted following an interview guideline, which consisted of 15 questions. An open structure of the guideline allowed deviating topics to emerge. The interview questions were based on research regarding work-related influences on health and well-being in the humanitarian field [6,9,21,27,28,29,30] and had a special emphasis on organisational factors. Topics included the daily work experience, organisational support, safety and security, prior experience in the humanitarian field and the living situation. The interview guideline was previously piloted with two volunteers who had been working in the humanitarian aid on Lesvos for several months. The interview guideline was adapted accordingly. A female M.D. candidate, who was residing on Lesvos during the time of the interviews, conducted all interviews. In the sampling process, efforts were made to not include interview subjects with a personal connection to the researcher. The interviews were held at a quiet, private space. All interviews were audio-recorded and lasted an average of 39 min (range 22–58 min). The majority of the interviews were conducted in English. Only one interview was conducted in German, as the interviewee and interviewer were both German native speakers.

### 2.4. Data Analysis

The interviews were transcribed and anonymised. Qualitative inductive content analysis [31] was applied to analyse the transcripts by using the software MaxQDA2020. This approach enables a systematic and theory-based extraction of the main content [32]. The coding of the transcripts regarding burdening and protective organisational factors was carried out by three investigators and in various work cycles. During the analysis, several feedback loops were completed to ensure a high quality and interpersonal credibility of the category system. Further, an analysis of gender-specific aspects in the category system took place. This analysis revealed categories that emerged only from interviews with one of the gender groups. To minimise a random accumulation of only seemingly gender-related factors, it was decided to only include categories that originated from at least three interviews of one gender group.

## 3. Findings

### 3.1. Overview

Burdening and protective factors of volunteers in the humanitarian aid were identified in organisational aspects of the work. In some cases, areas exhibit both burdening as well as protective factors. Accordingly, burdening and protective factors can be found in work procedures, team interactions and organisational support. Meanwhile, gender-specific disadvantages are only associated with burdening factors and the areas of joyful experiences and pleasant living arrangements contribute to protective factors only. The findings will be presented according to the research questions.

### 3.2. Burdening Organisational Factors in the Work

The qualitative content analysis of the interview transcripts revealed burdening organisational factors in three different categories. An overview over these factors is presented in Table 1.

#### 3.2.1. Burdening Work Procedures

In this study, work procedures compile all aspects associated with the execution of work tasks. The analysis identified several work procedures that led to an increased burden among the volunteers. Many of the interviewees named the performance of physically and mentally demanding work as a reason for stress and decreased subjective well-being. This included long working hours and hard physical labour. In addition, the feeling that individual skills were not considered or were overlooked by the organisation caused frustration among some interviewees. Furthermore, many of the volunteers felt emotionally burdened through witnessing the circumstances in the camp or interacting with camp residents. These experiences were sometimes related to pressure to fulfil conflicting roles, e.g., being friendly and open towards the refugees and completing a complex task under time pressure.
*“Keeping the balance between giving your everything and being happy all the time and also working […] so that’s sometimes giving me stress or disappointment in myself. That’s an internal struggle you have sometimes.”*(P. 3/female/age 25)

Further aspects concern the weekly work structure. The main challenges in this context were an inadequate briefing structure and poorly planned work schedules. One interviewee explained that:
*“[the coordinator] doesn’t really get it is exhausting to have shift off, shift on, shift off, shift on and I don’t think she always understands this when you say this to her”*.(P. 2/female/age 29)

Furthermore, several volunteers found that difficulties arose from an insufficient initial training and orientation to the work. To a lesser extent, volunteers felt negatively impacted by planning insecurities.

#### 3.2.2. Difficult Team Dynamics

Most NGOs operate as a larger team, which contributed to a variety of issues in the everyday work. Many interviewees named a fragmentary loop of information as a struggle they were facing. According to the interviewees, this lack of information hampered the completion of tasks and, thus, led to further struggles, e.g., receiving negative feedback. Furthermore, disharmonious interactions between different levels of the hierarchy caused further issues for some interviewees. These problems developed when the hierarchy was too rigid or unclear and were often associated with poor communication.

Other problems developed due to conflicts with the NGO management. In these cases, the decisions of the NGO management contradicted the interviewee’s experience. This conflict became especially relevant when external events affected the work. One interviewee explained how his organisation handled a particularly challenging situation:
*“[…] but when there was 12,000 people on the streets for two weeks, every single human being that was here should have helped. We shouldn’t spend three hours in a meeting thinking about this and that. This was a major frustration for me.”*(P. 12/male/age 33)

Moreover, the analysis identified a dysfunctional team as a cause of discomfort and decreased well-being. Reasons for the dysfunction were disagreements in the team or unsuitable volunteers:
*“[…] for me it’s really important to really try to get the best goal or something possible. If somebody is not doing that, that’s really making me angry. I actually have a lot of bad memories about working here and they are actually all related to this one person that cannot step up and do the work.”*(P. 16/female/age 24)

#### 3.2.3. Insufficient Organisational Support

Unhelpful or missing support mechanisms deprived volunteers of the opportunity to cope with stress arising from the work in a structured way. Several volunteers explained that the support systems did not meet their personal needs or that there was not enough support:
*“I feel like for some things we do experience at work, we actually need psychosocial support. […] I really feel it’s not enough because those peoples that offered help are coordinators. I really feel this is not enough and not just for me but for all the people.”*(P. 13/female/age 25)

Furthermore, some volunteers perceived a barrier to contact available support.

### 3.3. Protective Organisational Factors in the Work

Within the organisational structures, protective factors appear in five categories. Partly, these areas can be seen as contrasting expressions of the burdening organisational categories. Besides burdening work procedures, there were also constructive work procedures. Insufficient organisational support mechanisms can be opposed by helpful organisational support and difficult team interactions by favourable team interactions. Additionally, joyful experiences during the work and pleasant living arrangements were found to have a positive impact on the volunteers. Table 2 gives an overview of the protective organisational factors.

#### 3.3.1. Constructive Work Procedures

During the work, well-planned procedures were seen as protective by the volunteers and allowed them to find joy in their tasks. Many participants deemed it beneficial to have an extensive initial training and orientation to the work. In this way, volunteers felt more comfortable entering a new field of work and had the chance to address issues up front. During work, most interviewees enjoyed the experience of continuous learning and interacting with camp residents:
*“So, I love working closely with community volunteers and not just working with people but just being around another culture all the time. It makes you analyse your own culture and your own decisions, and this is a very important part of human growth in my opinion.”*(P. 12/male/age 33)

As this study identifies all aspects as work procedures that are related to the execution of work-related tasks, the co-ordination of assigned tasks is an important aspect for this category. Herein, some interviewees considered it beneficial when work in the camp was carried out as a team, ensuring that the pressure and responsibility was not only focused on one person. On an individual level, sufficient time to relax was important and achieved by clearly scheduled free times. Several volunteers explained that they were content because their work did cause them only little or no stress:
*“So, for me […] I can’t remember that I have been in a situation where I have felt very stressed.”*(P. 18/male/age 65)

Moreover, interviewees highlighted autonomy in the organisation of one’s own work and working in line with the personal skills and preferences as beneficial features of their work. One volunteer mentioned that opportunities to give feedback in the form of regular short debriefings had a positive effect.

#### 3.3.2. Favourable Team Interactions

Contentment among the volunteers increased when teams created a safe and pleasant atmosphere. In order to achieve this pleasant atmosphere, many named an open space for communication as an important factor. Moreover, a positive impression of the hierarchy in the NGO contributed to a good experience. This evaluation of the hierarchy was independent of its rigidity. Some felt supported through a clear structure and others appreciated a flat or flexible hierarchy. Further protective aspects included the work in a small team and collaborative team decision making:
*“There is no decision made without the entire team’s input. This for me is amazing. Even for the new people, it’s nice to feel involved in the decision process from the start.”*(P. 12/male/age 33)

#### 3.3.3. Helpful Organisational Support 

Different forms of support by the NGOs assisted the volunteers in coping with struggles while working in the refugee camp. Most of the interviewees considered peer support as one of the biggest forms of support:
*“You make friends here; you get to know people and then you share your thoughts with people you are driving to shift with or with your roommates […]. I think it’s more like an informal support system which works the best for me.”*(P. 7/female/age 29)

This kind of support was often related to living together or spending time with colleagues apart from the work. Material support, such as travel expenses or prepared food, were also an asset that some volunteers received. Less frequently mentioned, but still perceived positively, was the guidance through a co-ordinator, e.g., the opportunity to talk with someone with field experience and knowledge of the circumstances of the work. The few volunteers that received professional assistance, for example, through a psychologist, valued this experience. Few volunteers positively mentioned the guidance through strong values of the NGO and well-managed administration.

#### 3.3.4. Pleasant Living Arrangements

By arranging communal living, NGOs provided an important factor that many of the participants recounted. One participant highlighted the importance of his living situation in the following way:
*“I think we have the perfect environment to process it and to not be mentally exhausted by it. We get to come home to a warm house, which is an NGO house, so something we don’t even have to take care of. […] When we come home, […] we have our small techniques and mechanisms to create our home bubble and it’s a safe place and I feel really safe here.”*.(P. 10/male/age 23)

Those living arrangements showed to be important for many interviewees. Often, the interviewees reported a connection to the creation of positive relationships and the experience of peer support. These aspects also appear in other categories of protective factors.

#### 3.3.5. Joyful Experiences

Positively emphasised events during the work were an important resource for almost all interviewed volunteers that increased their motivation and strength to continue to work in the humanitarian field. The analysis identified two major categories for this area. Firstly, participants enjoyed their work and the positive impact it had, as described by one participant:
*“Your probing questions made me realize like my mental health has probably been way better because I am feeling like I’m acting in life, doing something that I enjoy doing rather than sitting in a hot office in the middle of November, answering emails which make no difference in the end of the day.”*(P. 5/male/age 28)

Secondly, creating positive personal relationships with colleagues, the beneficiaries or other people of the community was an important factor for the well-being of the interviewees. Positive relationships were often described together with the experience of strong peer support and, therefore, valued on multiple levels.

### 3.4. Gender-Specific Aspects within the Work

In the course of the analysis, several gender-specific aspects became visible. These aspects manifest as categories that exclusively emerged from interviews with either male or female volunteers. Table 3 shows an overview of these aspects. Many of the male volunteers described experiencing little or no stress at work, which they attributed to a lower workload compared to what they were used to or what they observed in other volunteers. None of the female volunteers gave a comparable description of their work situation. Only female volunteers characterised good work procedures as taking responsibility for a specific task or project and working in line with their personal preferences. On the other hand, unapproachable co-ordinators or the fear of judgement by co-ordinators led to an increase in pressure during the work among some female interviewees. It became apparent that more female volunteers struggled with the organisational support of their NGO, as all accounts for insufficient support mechanisms were made by women. 

Additionally, the issue of gender-based disadvantages caused an extra strain for the well-being of female volunteers in the refugee camp. Many of the female interviewees described one or several events in which they experienced difficulties or disadvantages that were directly related to their gender:
*“[…] I feel like I am not allowed to do the same kind of things my male counterpart could do because he is a man.”*(P. 1/female/age 27)

These kinds of different possibilities for men in the work environment were often justified with safety concerns. Other themes were difficult working conditions, such as a “*masculine culture*” in which *“[men] needed to prove themselves*” (P. 9/female/age 31) and unwanted advances which were “*intimidating*” (P. 2/female/age 29).

## 4. Discussion

This study investigated burdening and protective organisational factors for health and well-being of international volunteers in the humanitarian aid. Even though the volunteers in this study experienced joy in their work, there were still many aspects in the organisational structures that felt burdening and caused hardship in their everyday life. Some of these burdening experiences had a direct counterpart in the protective factors in the same area. Additionally, this study revealed gender-specific differences in the experience of burdening and protective organisational aspects. The findings will be discussed according to the main areas of the category system.

Several burdening and protective work procedures were identified and often revealed directly opposing experiences. For example, on the one hand, volunteers appreciated working in line with the personal skill set and, on the other hand, felt burdened by the feeling that individual skills were not considered. When considering the different impacts work procedures can have, it becomes evident that the NGOs can directly influence how their volunteers feel by implementing more protective and less burdening work procedures. Most previous research focused primarily on burdening factors in humanitarian work procedures, e.g., working long hours or having a high workload [9,17,18,20,33], but little is known about factors that have a positive impact. In this area, the only more frequently investigated topic is preparation and training for professional HAWs before the mission [15,33,34,35]. By revealing work procedures with a protective impact on the volunteers, this study makes an important contribution to narrowing this research gap.

However, not all studied factors have such clear implications. Interactions with the camp residents appear in the categories of burdening as well as protective work procedures. Inconclusive literature on this topic accompanies this observation. While research among disaster responders found that working directly with beneficiaries had a negative impact [36], a study among professional HAWs found a positive impact [13]. No research on the impact of this experience on international volunteers in the humanitarian aid was identified in the literature. The complexity of this experience may be partly explained by other, associated aspects. On the one hand, interacting with camp residents often occurred together with witnessing the circumstances in the camp or experiencing role ambiguity, which are both aspects that have been named as burdening by the interviewees. On the other hand, the feeling of doing good and creating positive relationships often occurred when interacting with the beneficiaries and was a major motivation and resource for the interviewees. As demonstrated by this example, a deep inspection into the various aspects of humanitarian work is necessary to understand the complex reality of volunteers working in this field.

The investigation of team interactions revealed many important new insights, as barely any scientific knowledge exists on the impact of team interactions among volunteers in the humanitarian aid. Some of the burdening team interactions, e.g., a fragmentary loop of information or unapproachable co-ordinators, have a protective counterpart, such as an open space for communication. As known from previous research in the humanitarian work, organisations can improve communication, for example, through team building measures [35] which would also improve the resilience of volunteers [22]. Additionally, this study found that the selection of suitable volunteers is important to prevent conflicts or dissatisfaction at a later stage. This finding underscores the results of previous research among professional HAWs [33,37,38].

Previous research often emphasised the importance of social or organisational support to mitigate adverse mental health outcomes among professional and volunteer workers in the humanitarian work [5,6,12,18,21,22,39,40]. In this study, peer support as a form of perceived organisational support posed an important resource for the interviewees. This finding supports previous research on professional HAWs and humanitarian volunteers [6,18,21]. Additionally, the interviewees valued professional support offers but sometimes distinct barriers kept volunteers from claiming them. This underscores recent research among professional HAWs, which found that many barriers to accessing psychosocial support link directly to the employing organisations [41].

Few previous studies found that poor living conditions were a cause for stress or unpleasant experiences among HAWs and humanitarian volunteers [18,42,43]. In this study, however, the protective effects of communal living arrangements by the NGOs were of far greater importance. No other studies on the positive effects of living arrangements for humanitarian volunteers were identified and, therefore, this study makes an important contribution to the current body of research.

Sporadically, past research stressed the need for a gendered approach in the humanitarian aid [44,45]. So far, there are no studies that investigate gender-based differences in organisational aspects among volunteers in the humanitarian aid. This study did not identify any gender-specific issues for male volunteers. However, research in different populations showed that men are more likely to show adverse attitudes towards seeking psychological help [46,47] and may, therefore, be less likely to report certain challenges. On the other hand, this study identified challenges for women in team interactions and in the provided organisational support. Additionally, female volunteers found themselves subject to gender-based discrimination and perceived this as an additional burden. Some of the events in which women experienced discrimination may be related to the fact that humanitarian aid is often provided in places or communities with more traditional gender roles. It is known from previous research that a stronger traditionality of gender roles can negatively impact the mental health of women [48], so this may also affect the humanitarian workers that engage in this environment. However, some findings of this study clearly relate to gender-based issues that originate from the organisation itself, such as a masculine work culture. This underscores recent findings regarding a lack of attention by humanitarian organisations towards gender-based issues inside of the organisations [49]. These findings suggest that health and well-being of volunteers is partly related to gender-specific organisational aspects and may, therefore, indicate a potential relation to the increased morbidity among female volunteers in the humanitarian aid [9,19].

So far, it became apparent that humanitarian work relates to many risks and hardships, which makes it important to understand the motives for continuous voluntary service in the humanitarian aid. Most of the interviewees described joy within their work and the creation of positive relationships as important resources and their main motivation. The study of Jachens et al. also described field work as most rewarding among professional HAWs [6]. Accordingly, these are important aspects to consider for the planning organisation.

A few limitations to this study should be considered when regarding the findings. Even though efforts were made to include a variety of people, the sample solely consisted of white people. Subsequently, issues related to racism were not captured. However, it is very likely that this is another major topic in the complex framework of the humanitarian aid. These struggles should be addressed in future research. Due to the global COVID-19 pandemic, some aspects of this study may not directly transfer to different circumstances. For example, due to lockdown regulations less socialising and peer support was possible, which might have led to different experiences for the volunteers. Regarding the gender-specific issues investigated in this study, it should be noted that no non-binary, transgender or intersex individuals were included in this study. Accordingly, the realities of these populations remain unknown and need to be addressed in future research. Additionally, further factors, such as age or the level of education, may influence the outcome of the gender-specific analysis. Even though this study identified burdening and protective factors for humanitarian volunteers in a Greek refugee camp, these findings may not directly transfer to other populations or settings and should, therefore, be validated in future research.

## 5. Conclusions

This study showed that organisational burdening and protective factors for health and well-being of volunteers in the humanitarian aid exist in the areas of work procedures, team interactions, organisational support, living arrangements, gender-specific disadvantages and joyful experiences. Often, appropriate planning can prevent burdening aspects or even remodel them into protective ones. Accordingly, work procedures that allow autonomy, taking responsibility and continuous learning should be implemented. It is also crucial to provide a good initial training and orientation to the work and sufficient time to relax. Additionally, this study underlines the importance of constructive team interaction, appropriate living arrangements and a helpful and accessible organisational support in the humanitarian sector. A sensitive approach to gender-specific needs and disadvantages should be established. Furthermore, organisations in the humanitarian aid sector should give volunteers the opportunity to witness the impact of their work and foster relationships with other volunteers or the surrounding community.

## Figures and Tables

**Table 1 ijerph-19-08599-t001:** Burdening organisational factors.

Burdening Organisational Factors	
Burdening work procedures	Inadequate briefing structure, individual skills are not considered, insufficient initial training and orientation, interactions with camp residents, physically and mentally demanding work, planning insecurity, poorly planned work schedules, role ambiguity, too much bureaucracy, witnessing conditions during operations in the camp
Difficult team interactions	Conflicts with NGO management, differences in communicating, disagreements in the team, fragmentary loop of information, rigid hierarchy, unapproachable co-ordinators, unclear hierarchy, unjustified negative feedback, unsuitable other volunteers
Insufficient organisational support	Barrier to contact the support offers (e.g., to contact a counsellor), insufficient NGO structures

**Table 2 ijerph-19-08599-t002:** Protective organisational factors.

Protective Organisational Factors	
Constructive work procedures	Autonomy in the organisation of tasks, continuous learning, good initial training and orientation, interactions with camp residents, little stress at work, regular debriefings, taking responsibility, time to relax, working in a team, working in line with personal preferences, working in line with personal skills
Favourable team interactions	Collaborative team decision making, comprehension for individual needs, open communication space, pleasant atmosphere in the team, positive view on hierarchy, small team
Helpful organisational support	Administration well managed, guidance through a co-ordinator, material support, peer support, professional psychological assistance, strong NGO values
Pleasant living arrangements	Pleasant living arrangements
Joyful experiences	Creating positive relationships, joy within the work

**Table 3 ijerph-19-08599-t003:** Gender-specific aspects in the category system.

	Burdening Factors	Protective Factors
Exclusively among male volunteers	/	**Work procedures** Little stress at work
Exclusively among female volunteers	**Insufficient organisational support** Barrier to contact support Disinterest in problems of other volunteers Unapproachable co-ordinators **Gender-based disadvantages** Different possibilities for men Difficult working conditions for women Unwanted advances	**Work procedures** Taking responsibility Working in line with personal preferences

## Data Availability

The data presented in this study are available on request from the corresponding author. The data are not publicly available due to the protection of the privacy of the participants.

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
