# Peer review of "Burdening and Protective Organisational Factors among International Volunteers in Greek Refugee Camps—A Qualitative Study"

_ijerph, 2022, doi:10.3390/ijerph19148599_

Round 1

Reviewer 1 Report

Thank you for opportunity to review this paper; I have asthma/children's health related non-profit experience at local, state, regional and national levels in USA. Please see comments provided in attached PDF file. Please note most things highlighted once (light/yellow) were typically out of interest; things highlighted twice (dark/orange) directly relate to a comment and/or a suggested edit or revision to the manuscript for its resubmission.

Author Response

We thank the reviewer for helpful insights and suggestions to our manuscript. We addressed all points and hope that the changes made improved the manuscript accordingly. Thank you very much for your interest and your effort.

Please see the attachment for our detailed response.  

Reviewer 2 Report

This article contributes to enriching the literature related to burdening factors and protective factors among humanitarian aid 12 volunteers, from an organizational point of view.

INTRODUCTION - The introductory section is well exposed, presents fairly recent references and adequately promotes the object of investigation. Furthermore, the gap in the literature that the study helps to fill is well clarified. I would suggest to the authors to enrich the discourse by making readers understand the organizational role and duties of volunteers in managing the need of humanitarian aid.

MATERIALS AND METHODS - The data collection for this study took place between December 2020 and January 2021. Could the COVID-19 pandemic have affected participant recruitment and Findings? This aspect should probably be argued. In addition, some aspects relating to the description of the sample should be investigated. For example: What are the tasks and roles of participants in their organizations? What is their educational qualification and their family status? This information could be important since even in the literature section these demographic variables are referred to as being associated with the mental health of humanitarian volunteers. If this information is available, Table 1 should be enriched.

RESULTS

This section should be renamed to Findings as Results are associated with quantitative research.

Despite the abundant material (28 interviews lasting an average of 39 minutes), only the transcripts of a few participants and a few excerpts of interviews are reported. Could the data be enriched? In addition, I would suggest that the authors specify the age or age ranges of the participant under the interview texts shown.

DISCUSSION

The fact that excerpts from the interviews are also reported in this section makes the work confusing. It should be considered whether to insert the explanations relating to what is indicated by the literature directly in the findings. In this case, it should be included in the discussion section a high level summary, which contains a high level link between the theory, data and literature. The findings relating to gender-specific aspects could be discussed more precisely. The authors state that there is too little gender-specific research in humanitarian assistance to reach a sufficient conclusion. However, the literature on gender studies and gender constructions could help to better discuss these findings.

CONCLUSION

The Conclusion section is well exposed and summarizes the protective and burdening factors in an applicative way.

Author Response

(The authors gave the same response as above.)

Round 2

Reviewer 1 Report

Thank you for revising manuscript. I had 1st chance to re-review it  July 11. I re-read abstract. The revised abstract reads more clearly and with more accuracy in terms of what can be said from a qualitative interview-based analysis with a small sample size (n=22). The authors responded to the two major comments provided completely and appropriately. Indeed, they they added good clarifications in two (not one) other parts of the Introduction (at p.2 lines 54-55 and 78-81, not just latter as noted in response to reviewers). Then, I reviewed manuscript revisions per my initial review comments and suggestions. In the opinion of this reviewer, the authors have done a thorough job in revising their manuscript and have adequately addressed each of this reviewer's concerns. This reviewer appreciates the authors replaced one Table and one Figure with simple text summaries, and added two references and ages of participants at direct quotes/examples given.